# Effect of Bovine *MEF2A* Gene Expression on Proliferation and Apoptosis of Myoblast Cells

**DOI:** 10.3390/genes14071498

**Published:** 2023-07-22

**Authors:** Jinkui Sun, Yong Ruan, Jiali Xu, Pengfei Shi, Houqiang Xu

**Affiliations:** 1Key Laboratory of Animal Genetics, Breeding and Reproduction in the Plateau Mountainous Region, Ministry of Education, Guizhou University, Guiyang 550025, China; s1907205810@163.com (J.S.); yruan@gzu.edu.cn (Y.R.); xji1484209550@163.com (J.X.); pengshishi2019@163.com (P.S.); 2College of Animal Science, Guizhou University, Guiyang 550025, China

**Keywords:** *MEF2A*, myoblasts, proliferation, apoptosis

## Abstract

Myocyte enhancer factor 2A (*MEF2A*) is a member of the myocyte enhancer factor 2 family. *MEF2A* is widely distributed in various tissues and organs and participates in various physiological processes. This study aimed to investigate the effect of *MEF2A* expression on the proliferation and apoptosis of bovine myoblasts. CCK8, ELISA, cell cycle, and apoptosis analyses were conducted to assess cell status. In addition, the mRNA expression levels of genes associated with bovine myoblast proliferation and apoptosis were evaluated using RT-qPCR. The results showed that the upregulation of *MEF2A* mRNA promoted the proliferation rate of myoblasts, shortened the cycle process, and increased the anti-apoptotic rate. Furthermore, the RT-qPCR results showed that the upregulation of *MEF2A* mRNA significantly increased the cell proliferation factors *MyoD1* and *IGF1*, cell cycle factors *CDK2* and *CCNA2*, and the apoptotic factors *Bcl2* and *BAD* (*p* < 0.01). These results show that the *MEF2A* gene can positively regulate myoblast proliferation and anti-apoptosis, providing a basis for the analysis of the regulatory mechanism of the *MEF2A* gene on bovine growth and development.

## 1. Introduction

Muscle yield and the quality of livestock are key indicators of the quality of livestock products. The growth and development of skeletal muscle and genetic characteristics significantly influence muscle yield and the quality of livestock [1]. Meanwhile, the productive performance of livestock is determined by the proportion of skeletal muscle. The type and amount of protein within skeletal muscle fibres and the content of intramuscular fat also affect muscle tenderness [2,3]. In addition, skeletal muscle is crucial for body coordination, metabolism, and homeostasis in mammals and is associated with a strong regenerative capacity [4,5]. Myogenesis (myoblast proliferation, differentiation, and fusion) is crucial for skeletal muscle development, including embryonic development, postnatal growth, and muscle regeneration after injury [6].

Myocyte enhancer factor 2 (MEF2), belonging to the MADS transcription factor superfamily, can regulate muscle-specific gene expression [7]. The MEF2 protein was first identified in developing the skeletal muscle myotubes of vertebrates. The MEF2 protein can anchor A/T-rich DNA sequences in the muscle creatine kinase (MCK) gene promoter [8]. The MEF2 gene family has four members in vertebrates: *MEF2A*, *MEF2B*, *MEF2C*, and *MEF2D*. However, this family is a single gene in Drosophila and nematodes [9,10]. MEF2 family members have similar structures, containing the MADS-box region, MEF2 central structural domain, and C-terminal transcriptional activation region [11]. The MADS-box region mainly binds to abundant A/T DNA sequences, the MEF2 central structural domain enhances the binding of A/T sequences in the regulatory regions of target DNA, and the C-terminal region participates in many key transcriptional processes during cell growth and development. Furthermore, the structural differences among MEF2 family members are mainly found in the C-terminal transcriptional region [12,13]. MEF2 is involved in some biological processes, including musculogenesis, skeletal development, and neurological development [14]. *MEF2A* activates many muscle-specific, growth factor-induced, and stress-induced genes. Moreover, *MEF2A* participates in life processes, such as cell proliferation and differentiation, apoptosis, and morphological changes [15,16]. *MEF2A* is located downstream of the signalling pathway regulating the expression of musculogenic genes. *MEF2A* functions as a regulatory factor and a structural protein carrier. Moreover, *MEF2A* performs a transcriptional activity not carried out by any other factors in the family [17]. Clark et al. found that the *MEF2A* gene specifically binds to the Gtl2/Dio3 locus, upregulates transcriptional cofactor *CITED2*, and promotes cardiomyocyte proliferation [18]. The *MEF2A* gene is one of the first genes detected during muscle regeneration in mice. Liu et al. showed that toxin injection in mouse tibialis anterior muscle can significantly downregulate *MEF2A* and *MEF2C* on day 2, and significantly upregulate *MEF2A* during later muscle regeneration [19]. Zhou et al. found that interference with *MEF2A* expression can promote atherosclerotic lesions in apoE knockout mice [20]. Studies have also shown that *MEF2A* can regulate vascular endothelial cell migration and anti-apoptotic cell death [21]. Besides the key activating and regulatory role of *MEF2A* in myogenesis, *MEF2A* is also involved in lipid metabolic pathways and plays a role in various biological processes. Moreover, *MEF2A* regulates muscle tissue growth and development, depending on tissue type [22,23].

However, no study has evaluated *MEF2A* in large domestic animals. Furthermore, systematic *MEF2A* studies on cattle are lacking. Guanling cattle are a unique breed in Guizhou Province, China, which have the advantages of strong physique and disease resistance, but their growth is slow and meat yield is low. In this study, Guanling cattle were used as experimental subjects with the aim of investigating the effects of changes in *MEF2A* gene expression on the proliferation and apoptosis of bovine myoblasts. Therefore, this study can help to investigate the regulation mechanism of beef qualitative traits and provide basic theoretical references for bovine growth and development.

## 2. Materials and Methods

### 2.1. Ethical Approval

Animal experiments were approved by the Laboratory Animal Ethics of Guizhou University (No. EAE-GZU-2021-E023, Guiyang, China; 1 November 2021).

### 2.2. MEF2A Overexpression and Construction of Interference Vector

The full-length CDS region sequence of bovine *MEF2A* (NM_001083638.2) was amplified and ligated to the PMD-19T vector. The recombinant *MEF2A* vector and pEGFP-C1 vector were double-enzymatically digested using *EcoR* Ⅰ and *Sal* Ⅰ. The recombinant digest product was obtained to obtain the *MEF2A* overexpression vector and negative control (NC) was supplied by our laboratory. Four interfering group target sequences and one NC group sequence were set based on the sequence of the coding region of *MEF2A* and shRNA design principles. The vector with the best interfering effect was selected for further analysis. The primer sequences were synthesized by GEMA Ltd. (Shanghai, China). The detailed shRNA target sequences are shown in Table 1.

### 2.3. Cell Identification and Transfection

Myoblasts were isolated and cultured from three healthy 3-day-old Guanling calves. The calves were born in Guanling Cattle Industrial Park, Anshun (Guizhou, China) then slaughtered. The longest dorsal muscle tissue was obtained, cut, and digested with type II collagenase. The filtrate was centrifuged, and the supernatant was discarded. The cells were then resuspended in DMEM-F/12 (Gibco, San Diego, CA, USA) containing 12% foetal bovine serum and 1.5% penicillin-streptomycin. The purity of the myocytes was determined using immunofluorescence. α-actin (Bioss, Beijing, China) was used as primary antibody and Alexa Fluor594 (proteintech, Suzhou, China) as secondary antibody. Cells were inoculated into 6-well cell plates. Each vector was transfected into cells using Lipofectamine 3000 (Thermo Fisher Scientific, Waltham, MA, USA) transfection reagent when cell density reached about 80%. The cells were harvested after 48 h of transient transfection for subsequent assays, including CCK8, ELISA, RT-qPCR, flow cytometry, cell cycle, and apoptosis analyses.

### 2.4. Cell Proliferation Activity Assay

The effect of *MEF2A* overexpression and the expression interference on the proliferative capacity of adult myoblasts was examined using CCK8 reagent (APExBIO, Houston, TX, USA). Briefly, myoblasts were inoculated in 96-well plates and incubated at 37 °C in 5% CO_2_ for 0, 6, 12, 24, 48, and 72 h, followed by the addition of CCK8 reagent (10 μL) to each well. An enzyme marker (Thermo Fisher Scientific, Waltham, MA, USA) was used to detect absorbance at 450 nm.

### 2.5. ELISA for GH and INS

First, each vector was transfected into myoblasts for 48 h. The cells were washed with PBS, then 180 μL of RIPA lysis solution (Solarbio, Beijing, China) was added, and they were lysed on ice for 5 min. Total protein was collected by vortexing at 12,000 rpm for 15 min. Absorbance was measured at 450 nm using an enzyme marker (Thermo Fisher Scientific, Waltham, MA, USA) according to the instructions of the GH and INS kits. Linear regression equations were calculated for the standards based on OD values.

### 2.6. Flow Cytometry for Cycle Assay

Cells were cultured in 6-well plates, and each vector was transfected into myoblasts for 48 h. The cells were collected via centrifugation after trypsin digestion, followed by the addition of 2 mL of 70% ethanol, then fixed at 4 °C overnight. the fixed cells were centrifuged and the supernatant was discarded. The cells were washed with PBS, resuspended with 100 μL of RNase A, and placed in a 37 °C water bath for 30 min. Then, 500 μL of PI staining solution was added, and the sample was left to react for 30 min at 4 °C away from light. The DNA content of the different transfected and control groups was measured via flow cytometry (CytoFLEX, Beckman, Brea, CA, USA).

### 2.7. Flow Cytometry Assay for Apoptosis Detection

Cells were collected and prepared, as described above. The cells were centrifuged and washed with precooled PBS, resuspended with 100 μL binding buffer, and incubated with propidium iodide (PI) solution and FITC Annexin V (Thermo Fisher Scientific) at room temperature for 15 min while away from light. The samples were analysed via flow cytometry.

### 2.8. Real-Time Quantitative PCR

Total RNA was extracted using TRIZOL reagent (Solarbio, Beijing, China). The total RNA was reverse-transcribed using StarScrip II First Strand cDNA Kit (Genstar, Beijing, China) to obtain cDNA. The effects of *MEF2A* expression changes on the expression levels of cell cycle factors *CDK2* and *CCNA2*, apoptosis factors *Bcl2* and *BAD*, and cell growth factors *IGF1* and *MyoD1* were detected via RT-qPCR analysis. qRT-PCR primer information is shown in Table 2. The RT-qPCR program was run via Bio-Rad CFX96™ (Thermo Fisher Scientific, Waltham, MA, USA). The reaction system consists of 5 μL of SsoFASTTM EvaGreen^®^ SuperMix, 0.5 μL each of upstream and downstream primers, 0.5 μL of cDNA, and 4 μL of ddH_2_O. Each sample had three triplicates. The reaction program consisted of pre-denaturation at 95 °C for 30 s, denaturation at 95 °C for 5 s, and annealing for 5 s (39 cycles). *GAPDH* was used as the internal reference gene. Gene expression was calculated via the 2^−ΔΔ*C*t^ method.

### 2.9. Statistical Analysis

Data are expressed as mean ± SD of three biological replicates. One-way analysis of variance (ANOVA) and the independent t-test were used to assess differences between groups via SPSS 18.0 software (IBM SPSS Statistics 18, Inc., Chicago, IL, USA). Statistical significance was analysed at *p* < 0.05.

## 3. Results

### 3.1. Cell Purity

The cells were identified via indirect immunofluorescence (Figure 1). Myocytes appeared red when bound to the anti-α-actin, while the DAPI nuclear stain was blue. The cytoplasmic and nuclear staining overlapped perfectly, indicating that the cultured cells were myoblasts.

### 3.2. Carrier Efficiency

The expression of the *MEF2A* gene and the efficiency of the interfering vector were analysed using qRT-PCR. The gene expression of the overexpressed OE-MEF2A was about 100 times (*p* < 0.01) higher than that of the control OE-NC (Figure 2A). Moreover, the expression of sh-RNA3 significantly decreased in the interference group compared with the control sh-NC (Figure 2B)(interference efficiency; 94.66%) (*p* < 0.01). The sh-RNA3 vector (sh-MEF2A) was selected for subsequent experiments.

### 3.3. Cell Viability Assay

The CCK8 results showed that OE-MEF2A significantly promoted the proliferative activity of myogenic cells after 6 h compared with OE-NC (Figure 3A). However, sh-MEF2A significantly reduced the proliferative activity of myoblasts cells after 6 h onwards compared with sh-NC (Figure 3B).

### 3.4. GH and INS Analysis

The ELISA results showed that *MEF2A* overexpression slightly increased GH content (*p* < 0.01) and INS content in myoblasts (*p* > 0.05) compared with OE-NC (Figure 4A). However, *MEF2A* interference decreased GH and INS content in myoblasts (*p* < 0.01) (Figure 4B), suggesting that the *MEF2A* gene can promote cellular activity.

### 3.5. Cell Cycle and Apoptosis

The effect of *MEF2A* overexpression and interference on myoblast cell cycle and apoptosis was examined via flow cytometry. Compared with OE-NC, the number of OE-MEF2A myoblast cells was significantly reduced in the G1 phase (*p* < 0.05). Furthermore, the number of S-phase myoblast cells was significantly increased (*p* < 0.01). Furthermore, G2/M was not significantly changed in the S-phase (*p* > 0.05) (Figure 5A), while the apoptosis rate of myoblast cells was reduced (*p* < 0.01) (Figure 5B). Compared with the sh-NC group, the number of myofibroblasts in the sh-MEF2A group significantly increased (*p* < 0.05) in the G1 phase and significantly decreased (*p* < 0.01) in the S and G2/M phases (Figure 5C). Moreover, the apoptosis rate significantly increased after the suppression of the *MEF2A* gene (*p* < 0.01). Furthermore, the inhibition of the *MEF2A* gene significantly increased the apoptosis rate of myoblast cells (*p* < 0.01) (Figure 5D). These findings suggest that the *MEF2A* gene can promote the proliferation of adult myoblasts. Moreover, *MEF2A* can stabilize cell activity.

### 3.6. RT-qPCR

*MEF2A* overexpression significantly increased the expression of the *MyoD1*, *IGF1*, *CCNA2*, *CDK2*, and *Bcl2* genes (*p* < 0.01) and significantly decreased the expression of the *BAD* gene (*p* < 0.05) in myoblasts (Figure 6A). In contrast, *MEF2A* interference significantly downregulated the *MyoD1*, *IGF1*, *CCNA2*, and *Bcl2* genes in myoblast cells (*p* < 0.01), significantly reduced the expression of the *CDK2* gene (*p* < 0.05), and slightly upregulated the *BAD* gene (*p* > 0.05) (Figure 6B). These results further confirm the effect of the *MEF2A* gene on myoblast cells.

## 4. Discussion

The protein encoded by MEF2A can participate in various cellular processes as homo/heterodimers, including muscle development, cell growth control, and apoptosis. The proliferation and differentiation of myoblast cells are crucial in muscle tissue growth through cell proliferation and protein deposition [24]. *MEF2A* promotes myofibril formation and participates in the regeneration and differentiation of skeletal muscle stem cells and myoblasts cells [25]. Wang et al. reported that the *MEF2A*-MEG3/DIO3-PP2A axis promotes bovine muscle regeneration [26]. Chen et al. also demonstrated that *MEF2A* can promote postnatal muscle growth and development in goat kids. Moreover, *MEF2A* significantly increases muscle fibre diameter in the loin [27]. *CDK2* is a cell cycle regulator that promotes the transition from the G1 to the S phase by forming a complex with *CCNA2*, thereby regulating cell cycle progression [28]. In this study, *MEF2A* upregulation or downregulation changed *CDK2* and *CCNA2* expression. *MEF2A* overexpression significantly decreased the number of G1-phase cells and significantly increased the number of S-phase cells, thus greatly shortening the G1-S phase process. Moreover, *MEF2A* interference did not affect the G1-S phase process, indicating that *MEF2A* accelerates the cell cycle process, consistent with the study of Wang et al. [29].

*MyoD1* positively regulates skeletal muscle growth and development, thus affecting the development and repair of muscle tissue [30]. *IGF1* can play a synergistic role with growth hormone (GH) to accelerate cell proliferation and differentiation and other physiological functions [31]. Herein, the CCK8 results showed that *MEF2A* upregulation significantly increased the viability of myoblast cells. Other assays also showed that *MEF2A* upregulation increased the expression of the *MyoD1* and *IGF1* genes, while *MEF2A* inhibition decreased these expressions. Growth hormone promotes growth, cell division, and proliferation, thus accelerating metabolism [32]. Insulin can be a long-term regulator of feed intake in ruminants, promoting glucose uptake from the blood. The glucose is then converted to glycogen for storage in the liver and muscle [33]. In this study, ELISA showed that *MEF2A* overexpression significantly increased GH levels and slightly increased INS levels. Moreover, *MEF2A* inhibition significantly decreased both INS and GH levels.

Zhou et al. reported that siRNA inhibition of *MEF2A* expression in apoE−/− mice increased the levels of pro-inflammatory cytokines, such as IL-6, MCP-1, TNF-α, and MMP-8, and atherosclerosis in apoE−/−mice [20]. Lu et al. also found that MEF2A deletion increased the risk of haemorrhage, inflammation, and hypercoagulable state in mice [34]. Resveratrol can delay the senescence and apoptosis of vascular endothelial cells (VECs). Resveratrol can also significantly upregulate the expression of *MEF2A* in VECs, thereby preventing apoptosis-induced oxidative stress [35]. In this study, *MEF2A* upregulation increased the activity of myogenic cells and had an anti-apoptotic effect. RT-qPCR also showed that *MEF2A* overexpression inhibited the expression of the pro-apoptogenic gene *BAD* and increased the expression of the anti-apoptogenic gene *Bcl2*. Furthermore, *MEF2A* inhibition increased the rate of apoptosis. One study reported that *MEF2A* reduced the expression level of Hspb7 in mouse skeletal muscle, which was effective in preventing muscle atrophy [36]. However, studies have also shown that *MEF2A* overexpression can disrupt myoblast differentiation patterns and apoptosis [29].

In addition, several studies have shown that *MEF2A*, *MEF2C*, and *MEF2D* have synergistic roles in muscle growth and development, especially *MEF2A* [37,38]. Synder et al. reported that mice with an overall deletion of *MEF2A* are defective in terms of skeletal muscle regeneration [38]. Deletion-of-function mutations in *MEF2A*, or *MEF2A* knockout, can significantly cause a disease phenotype in mice [39,40], suggesting that *MEF2A* has an irreplaceable function. Wu et al. found that *MEF2A* is evolutionarily higher on the list than *MEF2C* through phylogenetic analysis [41]. These findings reveal the importance and specificity of the *MEF2A* gene among the MEF2 family members.

## 5. Conclusions

In conclusion, *MEF2A* can positively regulate the proliferation and anti-apoptosis of bovine myoblasts through the regulation of key genes of the cycle (*CDK2*, *CCNA2*), growth (*MyoD1*, *IGF1*), and apoptosis (*Bcl2*, *BAD*). These results provide a basic theoretical reference for analysing the mechanisms through which *MEF2A* regulates bovine growth and development.

## Figures and Tables

**Figure 1 genes-14-01498-f001:**
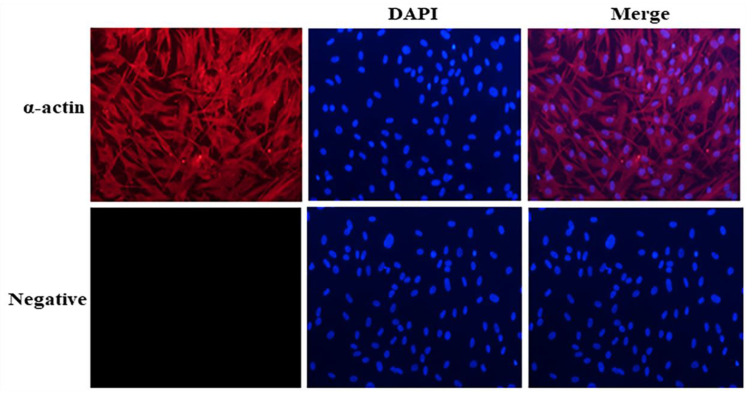
Indirect immunofluorescence detection of myoblasts bound to anti-α-actin (red, 1:100, Bioss, Beijing, China); nuclei stained with DAPI (blue) (200×).

**Figure 2 genes-14-01498-f002:**
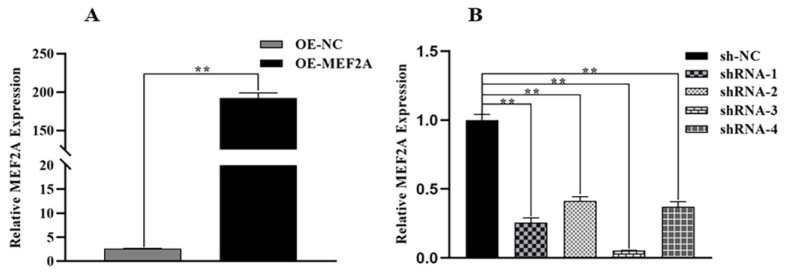
Detection of *MEF2A* expression and interference vector efficiency. (**A**) qRT-PCR analysis of *MEF2A* overexpressed vector and (**B**) *MEF2A* interference vector. Two asterisks (**) represent highly significant differences (*p* < 0.01).

**Figure 3 genes-14-01498-f003:**
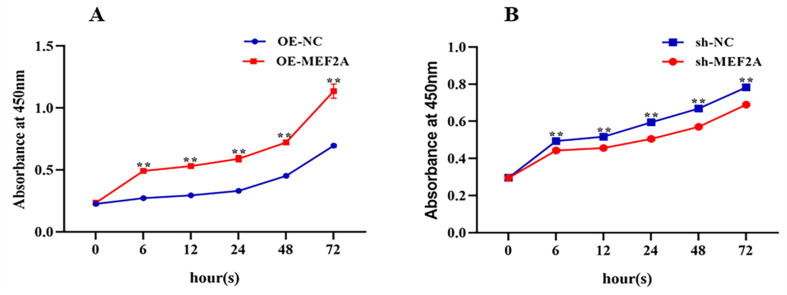
(**A**) CCK8 assay showing cell proliferation activity after transfection with OE-MEF2A and (**B**) sh-MEF2A. Two asterisks (**) represent highly significant differences (*p* < 0.01).

**Figure 4 genes-14-01498-f004:**
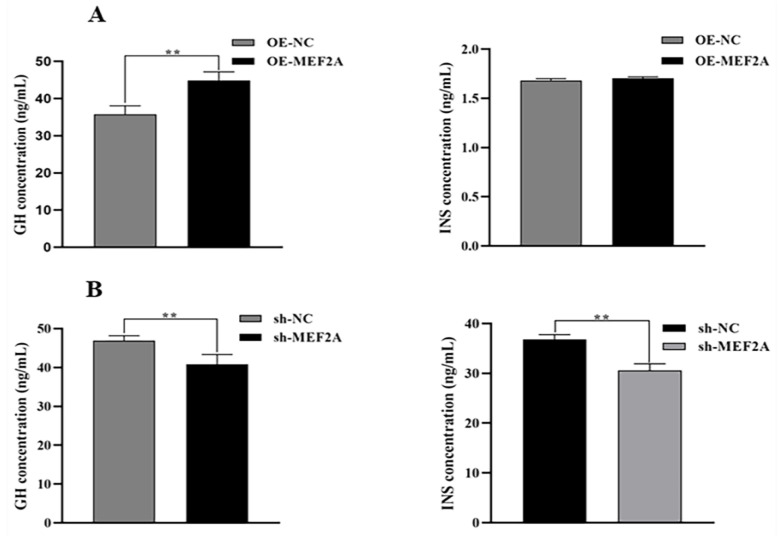
Effect of *MEF2A* expression on myoblast GH and INS. (**A**) ELISA showing the effect of OE-MEF2A transfection on the changes in GH and INS in myoblasts. (**B**) ELISA showing the effect of sh-MEF2A transfection on GH and INS changes in adult myoblasts. Two asterisks (**) represent highly significant differences (*p* < 0.01).

**Figure 5 genes-14-01498-f005:**
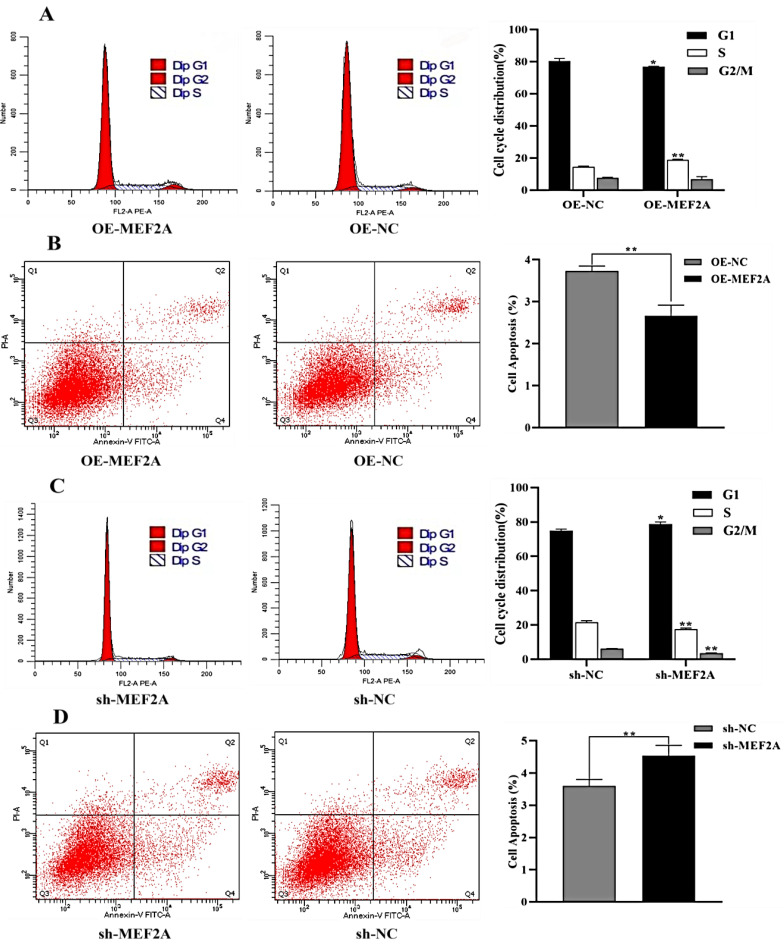
Detection of myoblast cycle and apoptosis via flow cytometry. (**A**) Characterization of myoblast cell cycle after OE-MEF2A transfection. (**B**) Characterization of apoptosis in myoblast cells after OE-MEF2A transfection. (**C**) Characterization of myoblast cell cycle after sh-MEF2A transfection. (**D**) Characterization of apoptosis in myoblasts cells through transfection with sh-MEF2A. One asterisk (*) indicates significant differences (*p* < 0.05). Two asterisks (**) represent highly significant differences (*p* < 0.01).

**Figure 6 genes-14-01498-f006:**
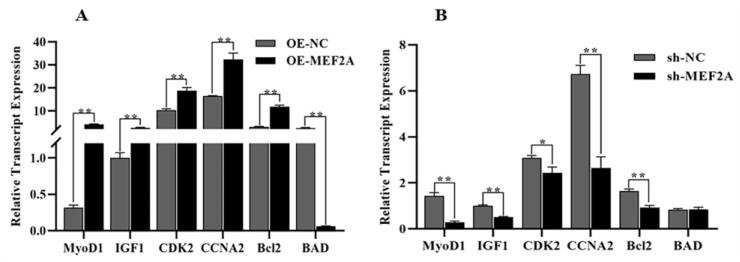
Effect of *MEF2A* expression on myoblast growth, proliferation, and apoptosis. (**A**) RT-qPCR showing the effect of OE-MEF2A transfection on the expression of *MyoD1*, *IGF1*, *CDK2*, *CCNA2*, *Bcl2* and *BAD*. (**B**) qRT-PCR showing the effect of sh-MEF2A transfection on the expression of *MyoD1*, *IGF1*, *CDK2*, *CCNA2*, *Bcl2*, and *BAD*. One asterisk (*) indicates a significant difference (*p* < 0.05), while two asterisks (**) indicate extremely significant differences (*p* < 0.01).

**Table 1 genes-14-01498-t001:** shRNA sequences.

Name	Sequence (5′-3′)
sh-NC	TTCTCCGAACGTCTCACGT
sh-RNA1	GCAGAACCAACTCGGATATTG
sh-RNA2	GCCTCCACTGAATACCCAAAG
sh-RNA3	GCAGCACCATTTAGGACAAGC
sh-RNA4	GCAGTTATCTCAGGGTTCAAA

**Table 2 genes-14-01498-t002:** Primer information.

Gene	Accession Numbers	Primer Sequence(5′-3′)	Product Size (bp)
*MEF2A*	NM_001083638.2	F: AATGAACCTCACGAAAGCAGAAC R: TTAGCACATAGGAAGTATCAGGGTC	106
*CDK*2	NM_001014934.1	F: CCTGGATGAAGATGGACG R: CTTGGAAGAAAGGGTGAG	101
*CCNA*2	NM_001075123.1	F: GCAGCCTTTCATTTAGCACTCT R: ATTGACTGTTGTGCGTGCTG	155
*Bcl*2	NM_001166486.1	F: ATGTGTGTGGAGAGCGTCAA R: ATACAGCTCCACAAAGGCGT	138
*BAD*	NM_001035459.2	F: TCCCAGAGTTTGAGCAGAGTG R: TTAGCCAGTGCTTGCTGAGAC	108
*MyoD*1	NM_001040478.2	F: AACCCCAACCCGATTTACC R: CACAACAGTTCCTTCGCCTCT	162
*IGF*1	NM_001077828.1	F: TGCGGAGACAGGGGCTTTTATTTC R: AAGCAGCACTCATCCACGATTCC	95
*GAPDH*	NM_001034034.2	F: TTGTGATGGGCGTGAACC R: GTCTTCTGGGTGGCAGTGAT	169

Note: F represents the upstream primer and R represents the downstream primer.

## Data Availability

The data presented in this study are available on request from the corresponding author.

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
