# Peer review of "Effect of Bovine MEF2A Gene Expression on Proliferation and Apoptosis of Myoblast Cells"

_genes, 2023, doi:10.3390/genes14071498_

Round 1

Reviewer 1 Report

The authors present a manuscript that is well founded. All methodologies are well described, tables and figures are well formulated. The manuscript is in line with the scope of the journal. However, I have an observation:

In line 16 of the abstract, the authors mention the qRT-PCR technique. I imagine it's the quantitative PCR after a reverse transcriptase reaction. It is usual to use the acronym RT-qPCR. Same thing on line 96 and other parts of the manuscript that are like that.

Correcting this, I think the manuscript is ready to be published as an article in this journal.

Reviewer 2 Report

Dear authors,

I will recommend your manuscript for publication, but I has some questions for edition.

L9 – “in various tissues and organs and participates in various” – wrong sentence. Concretize in your research sphere.

L11 –  “cell cycle” – is it analysis?

L66 - You started the “Introduction” with the importance of meat production in animal husbandry. Add this when formulating the purpose of the study.

L73 – “Myoblasts were isolated from healthy 3-day-old Guanling calves and cultured. The 73 calves were born in Guanling Cattle Industrial Park” – move it to another part. This is not ethical statement, this is object of study.

L73 – not clear – how many samples were used?

L91 – What is “1.5% penicillin-streptomycin”? Is it 15 gram per liter?

L92 – What were the cells for adhesion of myoblasts coated with?

L100 – “CCK8 reagent” – descript, and who is manufacturer?

L126 – Add manufacturer.

L166 – Not find this research in “Methods”

L198 – I didn’t find description of “GH and INS analysis”. If it is Grow Hormone, why this hypophysis hormone was synthesized in myoblasts?

L275 – Your “Discussion” is a literature review and a reiteration of research findings. You must compare your results with those of other authors. It is possible with results in other animals. And draw conclusions about how your study differs or confirms the results of other authors.

Regards,
